# Multi-Instance Learning Based Anomaly Detection Method for Sequence Data with Application to the Credit Card Delinquency Risk Control

## Abstract

Anomaly detection in sequence data is widely applicable to many fields and has significant commercial value to the financial industry. The focus of this paper is its utility as means to control credit card delinquency risk. Transactions that deviate from the typical data sequence are a common precursor of payment difficulty. Current detection methods do not effectively use transaction data to detect abnormal transactions. This makes it difficult to control the overdue payment risk. We propose a Multi-Instance Learning based anomaly detection (MILAD) method with well designed learning networks to address this problem. MILAD analyze users monthly transactions and payment history, and detect exceptions through well designed deep learning networks. By comparing the performance of MILAD and DAGMM, which is currently the most commonly used unsupervised deep learning algorithm for credit card risk control, MILAD best controls overdue risk by utilizing both transaction and payment information.

## 1 Introduction

In recent years, research on the anomaly detection of the sequential data has gradually become a hot topic. It has a very wide range of applications in many industry fields. Especially in the financial field, sequence data anomaly detection has a great commercial value. Traditional sequence data anomaly detection searches for the changes in several parameters of temporal data sequences, such as the time series data. For example, Gao et al. (2019; 2020) proposed methods to detect the anomaly position of the variance structure for the data sequence with smoothly changing mean function. Different from the traditional ones, this research focuses on analyzing the anomaly status of a multivariate time series data sequence by studying the influence of anomaly samples on the abnormal state of the whole data sequence in high dimensional space. The motivation of this paper is to the common credit default problem in the financial field. Effectively controlling the overdue risk of credit cards is a key issue. However, there is no effective algorithm which can effectively analyze the overdue risk by utilizing transaction samples in credit card transaction sequence so far.

For these overdue credit card users, most of their transactions are normal, and only a few transactions are abnormal, such as impulse purchase, fraudulent purchase, etc. These abnormal transactions are the main reasons to cause the overdue problem. However, current credit card overdue risk control approaches (Lucas & Jurgovsky, 2020; Chen & Guestrin, 2016; Liu et al., 2019; Bolton et al., 2001) having little power to utilize transaction information, and relying too much on business experience when conducting risk control, and being relatively cumbersome to use the model in practice, etc. A big challenge of utilizing these abnormal transactions is that there are no obvious post event features for assigning the abnormality labels to these anomaly transactions. The only label information we could use is the users monthly overdue information. Zong et al. (2018) proposed the DAGMM algorithm combining traditional unsupervised methods and deep auto encoders together, and achieved some good results. However, in practice, sample features are constructed artificially, which makes the representation of samples is not comprehensive enough. Therefore, the difference between abnormal samples and normal samples is limited, and the model can not distinguish them very well. Further more, this unsupervised algorithm cannot use the overdue information effectively.

At present, this method only has few applications in the cold-start businesses because of the absence of abnormal labels.

The characteristic of the credit card bill overdue risk detection is that the monthly bill has a label, but transactions on the bill do not have labels, which also happens in other application scenarios. The Multiple Instance Learning is a good solution to solve this kind of problem. There has been a lot of work done in this field, such as Carbonneau et al. (2018). Under the Multiple Instance Learning framework, samples are grouped into sets, which are defined as Bags. An abnormal status label is assigned to the entire bag. But no label is assigned to the samples in the bag. Then the relationship between the bag label and sample labels is determined based on the assumption of the Multiple Instance Learning. Ilse et al. (2018) proposed an Attention based the Multiple Instance Learning algorithm (ABMIL). ABMIL uses the Attention Neural Network to learn the attention weights of samples in a bag. Then the attention weights are used to aggregate samples in the bag, followed by the subsequent classification analysis. This aggregation method can assign weights to the samples in a bag, and then detect important samples based on sample weights. Inspired by their method, if we can utilize both individual transaction information and the overall bill overdue information simultaneously, we can improve the existing methods.

In this article, we propose a new anomaly detection algorithm based on the Multiple Instance Learning technique, named as Multiple Instance Learning for anomaly detection (MILAD). MILAD is a sequence sample information based anomaly detection method, which can make full use of sample information and sequence information. In the experiments studied in this paper, MILAD can control the overdue risk from the transaction perspective, which can provide more accurate and effective results. MILAD outperforms the most commonly used algorithms in terms of several major model evaluation criteria and provides a better performance in model interpretation.

The rest of the paper is organized as follows. The model and its proposed algorithm MILAD are introduced in Section 2, with the computation details of each module and their parameter optimization techniques. Section 3 is the experiment data analysis. In this section we conduct several experiments on the application data set and compare the results agaist those based on the DAGMM algorithm, which is the most commonly used method in the financial field. Section 4 is the summary of the paper.

## 2 METHODOLOGY

### 2.1 MODEL AND NOTATION

Suppose $X = \{\mathbf{x}_1, \ldots, \mathbf{x}_j, \ldots, \mathbf{x}_J\}$ is a time dependent multivariate sequence, where $\mathbf{x}_j \in \mathbb{R}^d$ is a sample of the sequence at time $j = 1, \ldots, J$. $y_j \in \{0, 1\}$ is the hiden label indicating the status of the sample $\mathbf{x}_j$, and 1 means abnormal. $y_j$ is the hidden state of the sample has to be predicted from the following model,

$$y_j = \begin{cases} 1 & \text{if} \quad f(x_j) \geq \delta \\ 0 & \text{else} \end{cases}, \quad j = 1, \ldots, J, \tag{1}$$

where $f : \mathbb{R}^d \to \mathbb{N}$ is a classifier based on feature mapping. We need to estimate the hidden state $y_j$ of each sample. $\delta$ is the threshold parameter discriminating the abnormal status of samples. An appropriate $\delta$ should be chosen according to the practical situation. Then the abnormal state label $Y$ of the sequence $X$ is modeled as

$$Y = \begin{cases} 1 & \text{if} \quad \mathscr{F}(y_1, \ldots, y_j, \ldots, y_J) \geq \Delta \\ 0 & \text{else} \end{cases}, \tag{2}$$

where $\mathscr{F} : \mathbb{R}^J \to \mathbb{R}$ is a function used to estimate the overall anomaly state of a data sequence. $\Delta$ is the threshold to discriminate the overall anomaly state of the data sequence.

Figure 1 is the flowchart of our entire modeling framework. The framework is composed of three parts. The first part is the Multiple Instance Learning based on the sample information and the

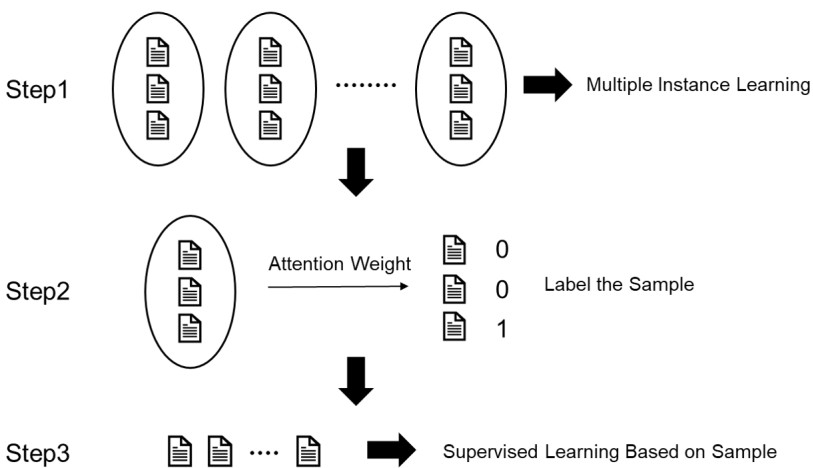

Figure 1: The framework of the MILAD algorithm

sequence information using the Attention mechanism. The second part is the anomaly label estimation of all samples in the data sequence according to the result from the previous Multiple Instance Learning procedure. The third part is the sequence anomaly detection procedure based on the estimated abnormal labels of samples using binary supervised learning method. Models are trained by the common optimization algorithm Adam (Kingma & Ba, 2014).

Algorithm 1 is the computational flow of our proposed sequence anomaly detection method MILAD. It is a Multiple Instance Learning based method, which can effectively associates the unknown sample label $y_j$ with the known sequence label $Y$ through the Attention network mechanism and achieve efficient modeling processes eventually. The MILAD algorithm constructs a risk analysis model $\mathscr{F}$ based on sample anomaly detection in a data sequence. In practice, taking the credit card overdue risk prediction businesses as an example, we can use the model $\mathscr{F}$ to evaluate card holders' overdue risk based on their transaction vector $\mathbf{x}' \in \mathbb{R}^d$. We can predict the overdue risk probability $p'$ through the model $\mathscr{F}$, and finally determine whether to intercept the transaction $\mathbf{x}'$ based on the actual needs of the business. In this way we can directly control the overdue risk in the transaction dimension. Comparing with traditional approach, controling overdue risk based on the MILAD algorithm is much more convenient in practice.

---

**Algorithm 1:** MILAD

**Input:** The Multi time series sample bag $X = \{\mathbf{x}_1, \ldots, \mathbf{x}_j \ldots, \mathbf{x}_J\}$ which is a collection of time series of length $J$, $\mathbf{x}_j \in \mathbb{R}^d$

**Step 1 Multiple Instance Learning**: Use the Algorithm 2 to estimate the classification probability $P = Sigmoid(W_C^\top Z + b)$ of the bag, the attention weight $w_j^* = \mathbf{w}_j / \sum_{m=1}^J \mathbf{w}_m$ of samples in the bag, and the abnormal probability $p_j = P w_j^*$ of samples in the bag.

**Step 2 Sample Anomaly Detection**: Based on the Multiple Instance Learning results from Step 1, detect the abnormal state of each sample $\mathbf{x}_j$ in the bag, and get the sample anomaly state set $S = \{\hat{y}_1, \hat{y}_2, \ldots, \hat{y}_J\}$.

**Step 3 Sequence Anomaly Detection**: Based on the abnormal sample detection results from the Step 2, use the classification (e.g., Xgboost) method to estimate The abnormal state $Y$ of the sample bag $X$.

**Output:** $Y$ the abnormal state of the sample bag, and $S$ the hidden abnormal state set.

---

## 2.2 MULTIPLE INSTANCE LEARNING

In model equation 1, the classification model $f$ is built upon the feature information of sequence samples in the bag. However, the anomaly state label $y_j$ of the sequence samples is generally unknown. Therefore, we cannot perform any supervised learnings directly. To effectively solve this problem, we use the Multiple Instance Learning approach. The Multiple Instance Learning model is composed of the following four parts: the transformer network $T$, attention network $W$, aggregation network $A$, and classification network $C$. The transformer network $T$ is designed to conduct a feature extraction and transformation on the original features. As mentioned in Foulds & Frank (2010), there are two different assumptions: the Standard Assumption and the Collective Assumption. The Standard Assumption is that each sample in the bag has its own label, the label of the bag is negative if all samples in the bag are negative, and the label of the bag is positive if there is at least one positive sample in the bag. The Collective Assumption states that the label of a bag cannot be determined by any single sample, but by the interactions between samples and the cumulative effect of some samples in the bag. Therefore, we propose two types of designs for the network $T$: the **Basic method** and the **Self-Attention based method**. The Basic method is adaptive to the standard assumption, while the Self-Attention based method is designed for the collective assumption, which has more practical usages. The attention network $W$ is used to learn attention weights of samples in the bag, and the attention weights are estimated through a module conducted by a two-layer gated neural network. The aggregation network $A$ is used to aggregate all samples in the bag. After calculating the attention weight of each sample through the attention network $W$, we can estimate the feature vector $Z_i$ of the bag by calculating the weighted sum of the sample vectors in the bag. The classification network $C$ is a network that classifies the bag vector. After the aggregation of samples in the bag, the classification problem is turned into a traditional binary supervised learning problem. To deal with features extracted from the neural network, a fully connection ($FC$) layer network together with the sigmoid activation function is used to calculate the anomaly classification probability $P_i$ of the $i^{th}$ bag. The details of the proposed multiple instance learning networks are listed in Appendix A.

---

**Algorithm 2:** The Multiple Instance Learning Algorithm

**Input:** The multi time series sample bag $X = \{\mathbf{x}_1, \mathbf{x}_2 \ldots, \mathbf{x}_J\}$

**Step 1** Randomly initialize the weights of the parameters in the $T, W, A, C$ network;

**Step 2** Transform the original sample through the network $T$ and obtain the transformed vector $\mathbf{h}_j = T(\mathbf{x}_j)$ through the Basic method or the Self-Attention method (Algorithm 3 in Appendix A).

**Step 3** Calculate the attention weight $w_j^*$ for samples in the bag through the network $W$.

**Step 4** Aggregate the samples through the network $A$: $Z = \sum_{j=1}^{J} w_j^* \mathbf{h}_j$.

**Step 5** Obtain the abnormal probability of the bag through the network $C$:
$P = Sigmoid(W_C^\top Z + b)$.

**Output:** The classification probability $P$ of the sample bag

---

Algorithm 2 is our designed Multiple Instance Learning Algorithm. We use the Attention mechanism to adaptively aggregate the samples in the bag $Y_i$. Since the feature extractor and classifier are conducted with neural networks, it allows to establish an end-to-end model to make the whole model to be more auto adaptive. Meanwhile, each step of the model is built upon neural networks, which makes the back propagation algorithm available for parameters optimization. All parameters are optimized by minimizing the Logarithmic loss

$$L(\Theta) = \frac{1}{N} \sum_{i=1}^{N} (Y_i \ln P_i + (1 - Y_i) \ln(1 - P_i)),$$

where $N$ is the sample size of the training data, $P_i$ is the anomaly probability of the $i^{th}$ bag.

## 2.3 Sample Anomaly Detection

After the Attention based Multiple Instance Learning, we obtain the probability $P_i$ of the label of the bag to be 1, and the attention weight $w_{ij}^*$ of each sample in the bag. Unlike the traditional Multiple Instance Learning, the estimated attention weights of the samples are more important here, which can be used to detect the key samples in the bag. That is, which sample in the bag has a significant impact on the abnormal status of the bag. The samples with larger attention weights have a greater impact on bags, and these samples are likely to be the key samples which lead to the bag abnormality. Therefore, we can combine the prediction results of the bag and the estimated attention weights together to predict the anomaly status of each samples in the bag. Let

$$p_j = P_i w_{ij}^*$$

be the probability of sample $\mathbf{x}_j$ being abnormal. The probability is used to rank samples in the bag, rather than discriminating samples with respect to the sample abnormality. By choosing a appropriate threshold $\delta$, the abnormal state of the sample is $y_{ij} = 1$ if $p_{ij} \geq \delta$.

## 2.4 Sequence Anomaly Detection

After using the Multiple Instance Learning for abnormal sample detection, we get the anomaly set $S = \{\hat{y}_1, \ldots, \hat{y}_J\}$, which contains the pseudo labels of all samples in the bag, and then we can do the binary supervised learning to estimate the sequence abnormal state $Y$ in model equation 2 using the classification approaches. In this paper, we adopt the Xgboost algorithm. However in practice, there are only a few sequences or sample bags which are abnormal. Therefore, the binary classification problem we are dealing with is a highly imbalanced data analysis problem. We should adopt the imbalance data analysis techniques. To evaluate the model performance for the imbalanced data, AUC will be a good choice.

# 3 Empirical Analysis

## 3.1 Data Preprocessing

Since payment data often contains sensitive private information about individuals or institutions, and only banks and other related institutions have access to it. Therefore the acquisition of such public data set is quite limited. The lack of available effective public datasets is also an challenge for researches in this field. In this work, we evaluate the performance of the proposed MILAD algorithm on a commonly used real data set, which is the Credit Card Fraud Detection (CCFD) (Dal Pozzolo et al., 2015) data. The CCFD data composed of transactions of credit card users in Europe on September 2013. This dataset includes 284,807 transactions, where 492 are abnormal transactions. It is a highly imbalanced dataset, which only has 0.17% of abnormal transactions. To deal with this highly imbalance problem of the data, we use the common undersampling method to sample 10% of normal transactions. Then the abnormal rate increases to 1.70%. Due to the privacy issues in this field, this dataset cannot provide the original transaction features and the user information. The data contains 28 principal component features, $\{V_1 \ldots V_{28}\}$, which are transformed from the original features, the transaction amount, and the anomaly label of each transaction.

In order to make the dataset suitable for solving our problem, we have to generate new dataset through the following data generating mechanism based on the original CCFD data. We randomly select a certain number of transactions from the original data set to form a sample bag, then take each sample bag as the user's transaction set $X_i$, and then label the bag according to the sample label in the bag. The labeling process of the sample bag is based on these two assumptions of the Multiple Instance Learning, which are the standard assumption and collective assumption. In the subsequent data analysis we assume that there are no available labels for the samples in the bag. Generating the dataset in this way can effectively mimic our desired scenario in which we have the label for user's transaction set, but lack of the labels for each transactions in the bag. The labeling rules for the sample bags are as follows. Under the Standard Assumption, as long as there is a sample $\mathbf{x}_{ij} \in \mathbb{R}^d$ whose label $y_{ij}$ is abnormal in the sequence set $X_i = \{\mathbf{x}_{i1}, \ldots, \mathbf{x}_{ij}, \ldots, \mathbf{x}_{iJ}\}$, the label of the overall sequence $Y_i = min\{1, \sum_{j=1}^J y_{ij}\}$. Under the Collective Assumption, only when the

sum of the amount of abnormal samples reaches a certain threshold, the label of the overall sequence $Y_i = 1$, if $\dfrac{\sum_{j=1}^{J} y_{ij}\nu_{ij}}{\sum_{j=1}^{J} \nu_{ij}} \geq \Delta$, where $\nu_{ij}$ is the transaction amount of the sample $\mathbf{x}_{ij} \in \mathbb{R}^d$, that is $\nu_j \in \{x_{j1}, \ldots, x_{jd}\}$. The parameter $\Delta \in (0, 1)$ is determined according to the specific application scenario.

For the convenience of the experiment, we assume the sample size in each bag is the same when generating the data set. Under the standard assumption, since we use the probability, which reflects the anomaly status of the sample, to rank the samples in the bag, rather than discriminating the samples. In the subsequent discrimination analysis, the threshold $\delta$ in model equation 1 is chosen to be the one which makes the highest $F1$ score in the training set. Under the collective assumption, we need to consider the proportion of the abnormal transaction amount among all transactions in the bag. When the proportion of abnormal transaction amount reaches the threshold $\Delta = 0.1$, we will consider the user's transaction bag to be overdue. Under each assumption, we have $N_1 = 200$ bags for training, $N_2 = 50$ bags for test. The size of the bag is $J = 10$. The anomaly rate of bags is $16.6\%$ for the standard assumption and $5.8\%$ for the collective assumption.

We show the performance of the proposed method on both the standard assumption and collective assumption. We first evaluate the sample anomaly detection performance, and then analyze the performance of the sequence anomaly detection. In the sample anomaly detection part, we compare MILAD with the most commonly used unsupervised anomaly detection algorithm DAGMM in the financial field in terms of common model evaluation criteria (Precision, Recall, F1 score, AUC), as well as the interpretability of these two methods. In the Sequence Anomaly Detection part, we first built an idealized model, which is a model constructed based on the ideal assumption that the hidden labels are all available, hereafter denoted by the **Ideal** model. We use the Ideal model as the benchmark since it always has the best performance among all possible methods. We use AUC to evaluate model performances.

The computational resources of our experiments are *Windows 10, Intel(R) Core(TM) i5-9300H, GeForce GTX 1650 GPU, 16GB Ram*. We use *Python 3.8* under *Tensorflow 2.5.0* enviroment.

### 3.1.1 SAMPLE ANOMALY DETECTION UNDER THE STANDARD AND COLLECTIVE ASSUMPTIONS

Table 1 is the network structures of the experiments. For DAGMM, we use the same network structure under both assumptions, and we also use the same hyperparameter settings in Zong et al. (2018) ($\lambda_1 = 0.1, \lambda_2 = 0.005$). For MILAD, the Basic method is adopted under the standard assumption, and the Self-Attention method is adopted under the collective assumption. $FC(a, b, c)$ is a full connection network, where $a$ and $b$ are the number of input and output neurons, $c$ is the activation function.

Table 1: Network structures under the standard and collective assumptions

| Method | Layer | Structure |
|---|---|---|
| DAGMM | Encoder | FC(28, 16, tanh) $\rightarrow$ FC(16, 4, tanh) $\rightarrow$ FC(4, 1, none) |
| | Decoder | FC(1, 4, tanh) $\rightarrow$ FC(4, 16, tanh) $\rightarrow$ FC(16, 28, none) |
| | Estimate Network | FC(3, 10, tanh) - Dropout(0.2) $\rightarrow$ FC(10, 2, Softmax) |
| MILAD (Standard) | Network T | FC(28, 16, ReLU) $\rightarrow$ FC(16, 8, ReLU) |
| | Network W | FC(8, 8, tanh) $\odot$ FC(8, 8, Sigmoid) $\rightarrow$ FC(8, 1) |
| | Network C | FC(8, 1, Sigmoid) |
| MILAD (Collective) | Network T | Refer to the structure in Algorithm 3, where $d = 8$ |
| | Network W | FC(8, 8, tanh) $\odot$ FC(8, 8, Sigmoid) $\rightarrow$ FC(8, 1) |
| | Network C | FC(8, 1, Sigmoid) |

Figure 2 shows the loss function curves of these two algorithms with respect to 1,000 epochs in the training processes. We can see that the DAGMM algorithm converges after 1,000 epochs. Therefore we choose the model after the 1,000 epochs of training as the final DAGMM model. For the MILAD algorithm, since we treat each bag as a sample group, the sample size is relatively small. It can be seen that the model is overfitted after 30 epochs of training under the standard assumption, and 10 epochs of training under the collective assumption. Therefore, under the standard assumption, we

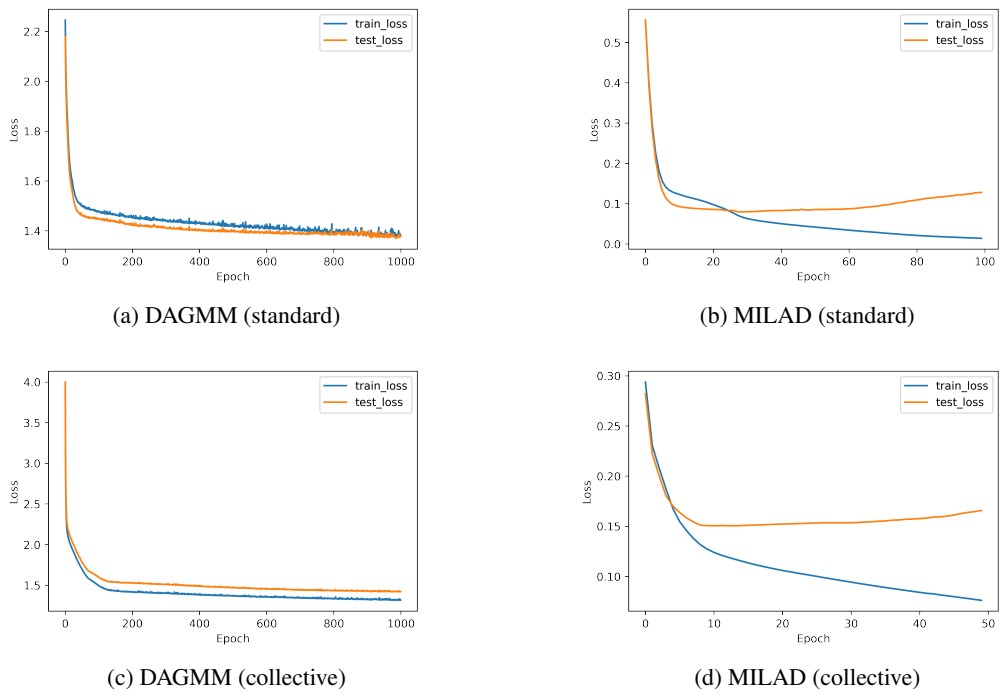

Figure 2: Loss curves under two assumptions

choose the model after 30 epochs of training as the final MILAD model, while under the collective assumption, we choose the model after 10 epochs of training as the final MILAD model.

The final output of the DAGMM model is the probability density of the sample. In order to compare it with the anomaly probability computed from the MILAD method, we use the function $f(x) = 1 - \frac{2}{\pi} \arctan(x)$ to convert it into the probability ranged in (0,1). Figure 3 is the ROC curve of the model trained by the DAGMM algorithm and the MILAD algorithm. It can be seen that the performance of the MILAD algorithm is significantly better than that of the DAGMM algorithm under both assumptions.

Table 2 is the comparison matrix in several common model evaluation criteria. It can be seen that MILAD is significantly better than DAGMM in terms of these common model evaluation criteria, such as Precision, Recall, F1 score and AUC. This is because DAGMM is an unsupervised learning method, while MILAD is a supervised learning algorithm, which can effectively utilize the label information of the bag for complex data through the attention based Multiple Instance Learning approach, and outperforms the unsupervised learning method. Therefore it is reasonable that MILAD achieves better performance, and is more useful in practice.

Table 2: Model comparison under two assumptions

| Assumption | Type | Method | Pecision | Recall | F1-score | AUC |
|---|---|---|---|---|---|---|
| Standard | Training | DAGMM | 0.0899 | 0.3722 | 0.1449 | 0.8397 |
| | | MILAD | **1.0000** | **0.8639** | **0.9270** | **0.9717** |
| | Test | DAGMM | 0.0971 | 0.4545 | 0.1600 | 0.8769 |
| | | MILAD | **0.9302** | **0.9091** | **0.9195** | **0.9627** |
| Collective | Training | DAGMM | 0.0580 | 0.2232 | 0.0921 | 0.7725 |
| | | MILAD | **0.6273** | **0.4000** | **0.4885** | **0.8854** |
| | Test | DAGMM | 0.0561 | 0.2526 | 0.0918 | 0.7856 |
| | | MILAD | **0.5781** | **0.3895** | **0.4654** | **0.8878** |

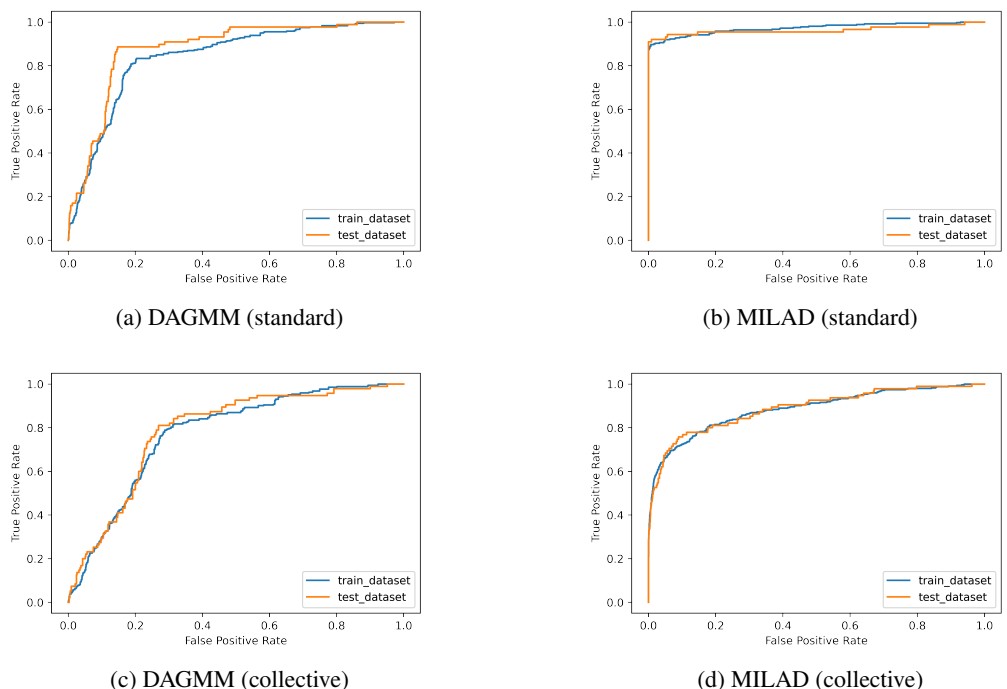

Figure 3: ROC curves under two assumptions

To show the interpretability of the MILAD algorithm, we also check these abnormal sample bags to see whether the method can identify the abnormal samples in the bag that cause the bag abnormality. Table 3 is the samples' anomaly state $(y_{ij})$ and their attention weights $(w_{ij}^*)$ of four $(i = 1, \ldots, 4)$ randomly selected anomaly sample bags under the standard and collective assumptions. The attention weights $\{0.75; 0.78; (0.30, 0.27); (0.20, 0.21)\}$ of the abnormal samples that cause the abnormality of the entire bag are significantly larger than other samples in the same bag. This result is consistent with our experiment setups, which fully demonstrates the outstanding interpretability of our MILAD method.

Table 3: Attention weights of two randomly selected cases under two assumptions

| Assumption | | $x_1$ | $x_2$ | $x_3$ | $x_4$ | $x_5$ | $x_6$ | $x_7$ | $x_8$ | $x_9$ | $x_{10}$ |
|---|---|---|---|---|---|---|---|---|---|---|---|
| Standard | $y_{1j}$ | 0 | 0 | 0 | 0 | 0 | **1** | 0 | 0 | 0 | 0 |
| | $w_{1j}^*$ | 0.03 | 0.04 | 0.02 | 0.00 | 0.02 | **0.75** | 0.05 | 0.02 | 0.04 | 0.03 |
| | $y_{2j}$ | 0 | 0 | 0 | **1** | 0 | 0 | 0 | 0 | 0 | 0 |
| | $w_{2j}^*$ | 0.03 | 0.01 | 0.03 | **0.78** | 0.00 | 0.00 | 0.00 | 0.01 | 0.10 | 0.03 |
| Collective | $y_{3j}$ | **1** | **1** | 0 | 0 | 0 | 0 | 0 | 0 | 0 | 0 |
| | $w_{3j}^*$ | **0.30** | **0.27** | 0.03 | 0.05 | 0.03 | 0.07 | 0.06 | 0.05 | 0.04 | 0.10 |
| | $y_{4j}$ | 0 | 0 | 0 | 0 | **1** | 0 | 0 | 0 | 0 | **1** |
| | $w_{4j}^*$ | 0.07 | 0.04 | 0.12 | 0.06 | **0.20** | 0.06 | 0.11 | 0.11 | 0.02 | **0.21** |

### 3.1.2 SEQUENCE ANOMALY DETECTION

For sequence anomaly detection we adopt the Xgboost algorithm, which is a commonly used binary supervised learning method in this field. All models are trained to achieve their best performances. The AUC results are shown in Table 4. It can be seen that in the absence of transaction labels, our MILAD algorithm still can achieve the similar performance as the Ideal model with respect to the AUC criterion, which is significantly better than DAGMM. We can conclude that MILAD is more feasible than DAGMM under both standard and collective assumption.

Table 4: AUC under the standard and collective assumptions

| Assumption | Type | Ideal | DAGMM | MILAD |
|---|---|---|---|---|
| Standard | Training | 1 | 1 | 1 |
| | Test | 0.98 | 0.87 | **0.98** |
| Collective | Training | 1 | 1 | 1 |
| | Test | 0.98 | 0.80 | **0.94** |

## 4 CONCLUSION

In this paper, we focus on the anomaly state evaluation of the data sequence caused by the abnormal samples contained in it. We propose a anomaly detection algorithm MILAD based on the Multiple Instance Learning techniques. We apply the proposed method to the delinquency risk detection in the credit card industry. The empirical results demonstrate that MILAD overcomes many short-comings that existing methods have through its use of the sample information and the sequence anomaly information simultaneously to effectively identify abnormal samples. The proposed method can help financial institutions to control the overdue risk based on transactions directly and effectively.

AUTHOR CONTRIBUTIONS

ACKNOWLEDGMENTS

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

300 APPENDIX

## A  MULTIPLE INSTANCE LEARNING NETWORK

302 In model equation 1, the classification model $f(\cdot)$ is built upon the feature information of sequence
303 samples in the bag. However, the anomaly state label $y_j$ of the sequence samples is generally
304 unknown. Therefore, we can not perform any supervised learnings directly. To effectively solve this
305 problem, we use the Multiple Instance Learning approach.

306 The Multiple Instance Learning model is composed of the following four parts: the Transformer
307 Network T, Attention Network W, Aggregation Network A, and Classification Network C. Accord-
308 ing to the structure of the Transformer Network, we have two types of designs: the Basic method
309 and the Self-Attention based method. The Basic method is adaptive to the Standard Assumption,
310 while the Self-Attention method is designed for the Collective Assumption, which has more prac-
311 tical usages. Figure 4 and Figure 5 are the network structures of the Multiple Instance Learning
312 model corresponding to the Basic method and the Self-Attention method.

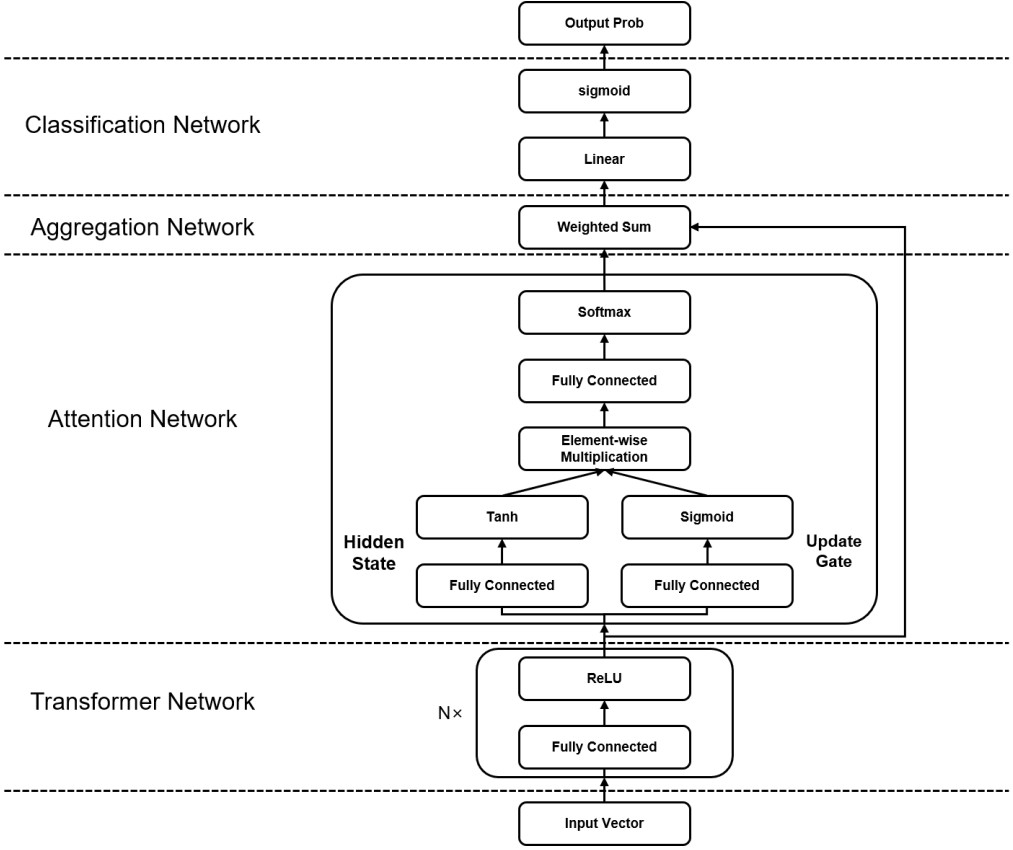

Figure 4: Multiple Instance Learning Network Structure (Basic Version)

313 The Algorithm 3 is the detailed transformation algorithm based on the Self-Attention method.

### A.0.1  TRANSFORMER NETWORK T

315 The function of the Transformer Network T is to conduct a feature extraction and transformation on
316 the original features. There are two approaches based on different assumptions: the Basic method
317 and the Self-Attention method. Figure 4 and Figure 5 are the network structures of the Multiple
318 Instance Learning model corresponding to the Basic method and the Self-Attention method.

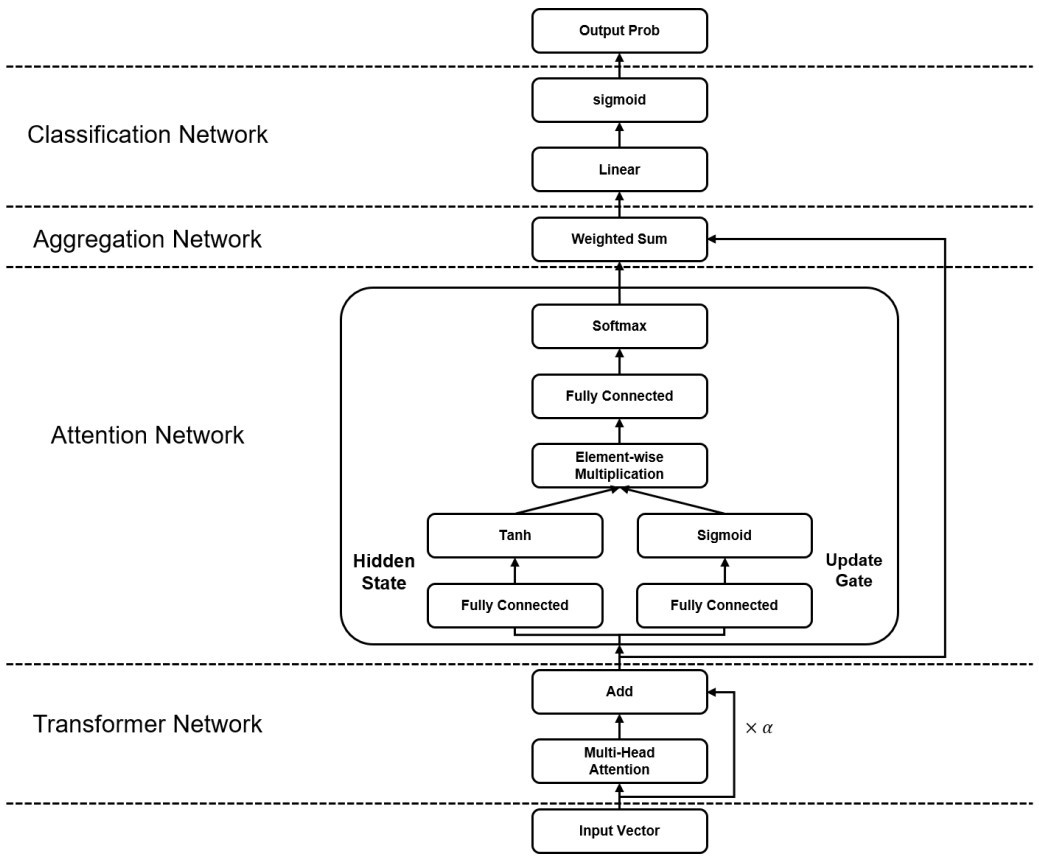

Figure 5: Multiple Instance Learning Network Structure (Self-Attention Version)

---

**Algorithm 3:** The Feature Transformation Algorithm Based on the Self-Attention Method

---

**Input:** The multi time series sample bag $X = \{\mathbf{x}_1, \ldots, \mathbf{x}_J\}$

**Step1**   Rewrite the input sample vectors in matrix form $X = [\mathbf{x}_1, \ldots, \mathbf{x}_J]^\top$;

**Step2**   Calculate $Q, K, V$ matrix:
$$Q = XW_Q$$
$$K = XW_K$$
$$V = XW_V$$

**Step3**   Calculate attention weight:

$$W_A = \mathrm{Softmax}(\frac{QK^T}{\sqrt{d_1}})$$

**Step4**   Calculate transformed matrix:

$$T = [\mathbf{t}_1, \ldots, \mathbf{t}_J]^\top = W_A V$$

**Step5**   Output the transformed feature vectors through Soft-Transoformer $\mathbf{h}_j$:

$$\mathbf{h}_j = \mathbf{x}_j + \alpha \mathbf{t}_j, \forall j = 1 \ldots J$$

**Output:** The transformed vectors $H = \{\mathbf{h}_1, \ldots, \mathbf{h}_J\}$

---

**The Basic Method**   As mentioned in Ilse et al. (2018), we use a double-layer neural network to calculate the attention weight. Let $\mathbf{x}_j$ be the sample feature vector, then $X = \{\mathbf{x}_1, \ldots, \mathbf{x}_J\}$ will be the corresponding sequence bag. The attention weight is

$$w_j^* = \frac{\exp\{W_2^\top \tanh(W_1^\top \mathbf{x}_j)\}}{\sum_{i=1}^{J} \exp\{W_2^\top \tanh(W_1^\top \mathbf{x}_i)\}}.$$

Then, the feature vector $Z \in \mathbb{R}^d$ for the bag will be estimated as the weighted average of the sample vectors. That is

$$Z = \sum_{j=1}^{J} w_j^* \mathbf{x}_j.$$

The advantages of this approach are all weight parameters are able to be optimized in the training process and the each sample in the bag has its own weight. Therefore, the feature vector conducted in this way has more interpretability. Meanwhile, we could also present a exception sample detection based on the their weights. The sample with a larger weight will have a larger chance to be a critical sample in the bag.

For the Basic approach, since it has been assumed that there is no interaction effect and no structural information between samples in the bag, sample $\mathbf{x}_j$ can be transformed into a feature vector $\mathbf{h}_j$ directly using a two-layer fully connected network,

$$\mathbf{h}_j = W_2^\top \sigma(W_1^\top \mathbf{x}_j + \mathbf{b}_1) + \mathbf{b}_2,$$

where $\sigma(\cdot)$ is the activation function.

**The Self-attention Method**   The Basic method assumes that the samples are independent when calculating the Attention mechanism. However, in practice, there exists various interaction effects between samples. In order to learn the interaction effectively, Vaswani et al. (2017) introduced a Transformer framework based on the Attention mechanism to obtain the interaction information between sequences composed of words. Based on this method, Rymarczyk et al. (2021) proposed a Soft-Transformer framework. The Soft-Transformer transforms samples into feature vectors first before the Attention based Multiple Instance Learning, which can explore the interactive information between samples more effectively. Let $\mathbf{x}_j$ be the sample vector, and $X = [\mathbf{x}_1, \ldots, \mathbf{x}_J]^\top$ be the sample matrix composed of vectors. Firstly, we use weight matrix $W_Q^{d \times d_1}, W_K^{d \times d_1}, W_V^{d \times d_2}$ to calculate the corresponding matrices $Q^{J \times d_1}(Query), K^{J \times d_1}(Key), V^{J \times d_2}(Value)$, where $Q = XW_Q$, $K = XW_K, V = XW_V$. We usually make $d_1 = d_2$. Then we have

$$W_A = \mathrm{Softmax}(\frac{QK^T}{\sqrt{d_1}}).$$

Finally we get the transformed sample matrix:

$$T = W_A V = \mathrm{Softmax}(\frac{QK^\top}{\sqrt{d_1}})XW_V,$$

where $T = [\mathbf{t}_1, \ldots, \mathbf{t}_J]^\top$ is a transformed sample matrix, and $\mathbf{t}_j$ is the transformed sample vector. Then we can perform the Multiple Instance Learning in the same way. Using this method we can get the transformed sample vector and the interaction information between samples. However, after transformation, the actual meaning of the vector is different from those for the original samples. The subsequent sample weights do not represent the importance of the samples anymore, and cannot be used to discriminate critical samples. Therefore, we use the Soft-Transformer to transform the output vector $\mathbf{t}_j$ into $\mathbf{x}_j + \alpha \mathbf{t}_j$. The Soft-Transformer we use makes the weight of the transformed vector still has the ability to reflect the importance of the samples after considering the interaction information in the analysis. In the subsequent analysis, we perform a critical sample discrimination based on attention weights. Figure 6 is the schematic diagram of the transformation process.

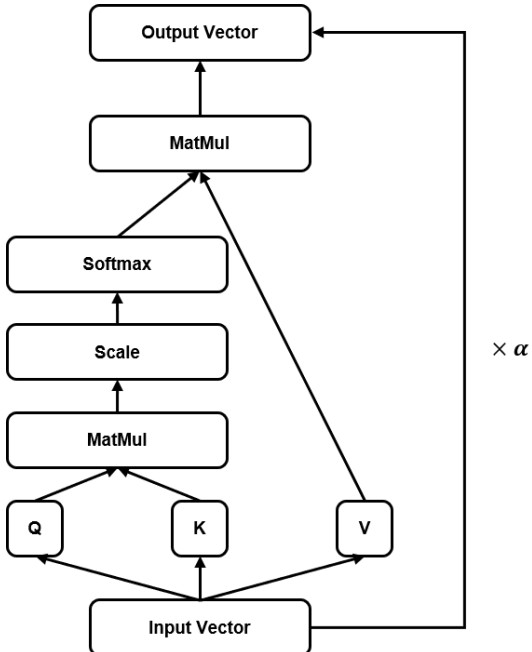

Figure 6: The structure of the Soft-Transformer

355 A.0.2 ATTENTION NETWORK W

The Attention Network is used to learn attention weights of samples in the bag, and the attention weights are estimated through a module conducted by a two-layer gated neural network:

$$\mathbf{v}_j = tanh(V^\top \mathbf{h}_j),$$

$$\mathbf{u}_j = Sigmoid(U^\top \mathbf{h}_j),$$

$$w_j = W_a^\top (\mathbf{v}_j \odot \mathbf{u}_j),$$

$$w_j^* = \frac{\exp\{w_j\}}{\sum\limits_{i=1}^{J} \exp\{w_i\}},$$

where $\mathbf{v}_j$ is the hidden state, $\mathbf{u}_j$ is the updated gate state, $\odot$ represents the element-wise multiplication of the vector, $w_j$ is the attention weight, and $w_j^*$ is the normalized attention weight by Softmax.

A.0.3 AGGREGATION NETWORK A

The Aggregation Network is used to aggregate all samples in the bag. After calculating the attention weight of each sample through the Attention Network $W$, we can estimate the feature vector $Z_i$ of the bag by calculating the weighted sum of the sample vectors in the bag,

$$Z_i = \sum_{j=1}^{J} w_j^* \mathbf{h}_j.$$

where $w_j^*$ is the attention weight of each sample in the bag, $\mathbf{h}_j$ is the feature vector obtained from the Transformer Network $T$.

### A.0.4 CLASSIFICATION NETWORK C

The Classification Network is a network that classifies the bag vector. After the aggregation of samples in the bag, the classification problem is turned into a traditional binary supervised learning problem. To deal with features extracted from the Neural Network, a fully connected (FC) layer network together with the Sigmoid activation function is used to calculate the anomaly classification probability $P_i$ of the $i^{th}$ bag, where

$$P_i = Sigmoid(W_C^\top Z_i + b).$$

