# OpenReview forum: "Multi-Instance Learning Based Anomaly Detection Method for Sequence Data with Application to the Credit Card Delinquency Risk Control"
_ICLR.cc/2024/Conference — ICLR 2024 Conference Withdrawn Submission_

### Official Review · Reviewer_eMfY · 2023-10-29

**Soundness:** 2 fair
**Presentation:** 2 fair
**Contribution:** 2 fair
**Rating:** 3
**Confidence:** 4

**Summary:**

Regarding the problem of credit card default, the paper designed a multi-instance anomaly detection method called MILAD, which achieved significant improvements compared to the common DAGMM on the CCFD dataset.

**Strengths:**

+ The paper builds on the multi-instance basis of the MILAD method and makes assumptions to make the prediction method more reasonable.
+ MILAD shows significant improvement compared to DAGMM on the CCFD dataset.

**Weaknesses:**

+ The paper's analysis of the field of credit card default prediction is insufficient.
+ The paper's experiments were only conducted on one dataset, CCFD, and the compared methods were too simplistic and outdated, which lacks persuasiveness.
+ The structure of the paper is somewhat unique, and it is not easy for readers to quickly understand the main innovations and contributions of the paper.

**Questions:**

The starting point of the paper is not problematic, but it seems to lack a holistic understanding of the field, and the experiments are somewhat weak, resembling a simple technical report.

---

> ### Author Response · Authors · 2023-11-16
>
> We are doing research on the anomaly detection of the sequential data in high dimensional space. We are focusing more on the methodology development for real wold cases. The real world unsupervised learning problems like the credit card risk control studies are very complicated. Currently there is no specific approach to this problem. DAGMM seems to be one of the most adaptable method so far. It combining traditional unsupervised methods and deep auto encoders together, and achieved some good results. However, in practice, sample features are constructed artificially, which makes the representation of samples is not comprehensive enough. Therefore, the difference between abnormal samples and normal samples is limited, and the model can not distinguish them very well. Further more, this unsupervised algorithm cannot use the overdue information effectively because of the absence of abnormal labels for transactions. MILAD are designed for solving the anomaly detection problems in these applications similar to the credit card risk control study.

---

> > ### Comment · Reviewer_eMfY · 2023-11-21
> >
> > Although the author further elaborated on the purpose of the paper, this reply did not solve my doubts well.

---

### Official Review · Reviewer_5JS1 · 2023-10-31

**Soundness:** 2 fair
**Presentation:** 2 fair
**Contribution:** 2 fair
**Rating:** 5
**Confidence:** 5

**Summary:**

The document presents a method called Multi-Instance Learning based Anomaly Detection (MILAD) for detecting anomalies in credit card transactions. The focus of the paper is on controlling credit card delinquency risk by identifying abnormal transactions. The existing methods do not effectively use transaction data to detect abnormal transactions, making it difficult to control the risk of overdue payments.
MILAD is designed to address this problem by analyzing users' monthly transactions and payment history using deep learning networks. It utilizes both transaction and payment information to detect exceptions. The performance of MILAD is compared with DAGMM, the most commonly used unsupervised deep learning algorithm for credit card risk control. The results show that MILAD effectively controls overdue risk by utilizing both transaction and payment information.

**Strengths:**

>The paper targets a very practical and significant area: credit card delinquency risk control.
>The authors have done a reasonable job highlighting the limitations and gaps in current approaches, such as the inability to effectively utilise transaction information and reliance on business experience, which adds value to the motivation of their work.
>The proposed algorithm effectively combines individual transaction information and overall bill overdue information, allowing for more accurate and effective anomaly detection.
>According to the empirical analysis in the paper, MILAD outperforms the DAGMM algorithm, the most commonly used unsupervised deep learning algorithm for credit card risk control.

**Weaknesses:**

>The paper mentions that 200 bags were used for training and 50 for testing, with the transactions forming a bag being randomly chosen. However, it's essential to calculate and report the standard deviation in addition to the mean for experimental results. This information helps assess the variability in the model's performance.
>While the paper mentions the computational resources used, there is no discussion of the actual resource utilization. Given that the algorithm employs transformer, attention, aggregation, and classification networks, it is important to address the computational expense associated with these components to ensure a comprehensive comparison with DAGMM.
>The paper mentions using AUC for evaluating model performance due to the highly imbalanced nature of sequence anomaly detection. However, the authors should consider using a more comprehensive set of evaluation metrics, including precision, recall, and F1-score. These metrics provide a more well-rounded view of the algorithm's performance, ensuring a complete evaluation.

How about datasets having only one fraudulent transaction? DAGMM requires at least two anomalies to be applicable. However, there are many real-world instances where there is one anomaly==fraudulent transaction. How does the method behave in such cases? Benchmark comparisons are not enough. For some reason, tree-based methods (dbTAI, MGBTAI) , Elliptic Envelope, Isolation Forest etc, have been ignored.

**Questions:**

Have the authors considered and implemented any regularization techniques to address the issue of overfitting observed in the experiments? Regularization methods, such as dropout or weight decay can help mitigate overfitting and improve model generalization.
>Has the MILAD algorithm been tested or analyzed for its generalizability on datasets and industries beyond credit card fraud detection? For example, exploring its effectiveness in stock trading scenarios where labels are available only for entire trading sessions, not individual trades, would be intriguing. Such experiments could showcase the algorithm's adaptability and utility in diverse contexts.

---

> ### Author Response · Authors · 2023-11-16
>
> Thank you for your valuable advise. MILAD does not have any requirement on the number of fraudulent transaction. The real world unsupervised learning problems like the credit card risk control studies are very complicated. Currently there is no specific approach to this problem. MILAD are designed for solving the anomaly detection problems in these applications similar to the credit card risk control study. Although MILAD has achieved some credits, it is still far away from SOTA. We are still working on the methodology development. We are working on the new MILAD frame work with regularization. We will give full evaluation with multiple criteria on multiple application data resources according to your advice for the updated MILAD approach in the future.

---

### Official Review · Reviewer_nXvi · 2023-10-31

**Soundness:** 2 fair
**Presentation:** 2 fair
**Contribution:** 2 fair
**Rating:** 6
**Confidence:** 5

**Summary:**

Anomaly detection in sequence data is widely applicable to many fields and has
significant commercial value to the financial industry. The focus of this paper is its
utility as means to control credit card delinquency risk. Transactions that deviate
from the typical data sequence are a common precursor of payment difficulty. Current detection methods do not effectively use transaction data to detect abnormal transactions. This makes it difficult to control the overdue payment risk. The authors propose a Multi-Instance Learning based anomaly detection (MILAD) method with
well designed learning networks to address this problem.

**Strengths:**

Paper is well formatted

Topic is interesting

The algorithms are interesting

**Weaknesses:**

References are dated for the topic

This area is well researched and authors fail to create a clear research gap

Readability can be improved

Appendices are hard to follow

**Questions:**

Why was DAGMM chosen for comparative analysis?

Were the other methods re-implemented or numbers were taken from the papers?

MILAD novelty is what exactly? How does it differ from other similar methods?

---

> ### Author Response · Authors · 2023-11-16
>
> To questions:
> 1. DAGMM is the most commonly used method with convincible performance in this area so far.
>
> 2&3. The real world unsupervised learning problems like the credit card risk control studies are very complicated. Currently there is no specific approach to this problem. DAGMM seems to be one of the most adaptable method so far. It combining traditional unsupervised methods and deep auto encoders together, and achieved some good results. However, in practice, sample features are constructed artificially, which makes the representation of samples is not comprehensive enough. Therefore, the difference between abnormal samples and normal samples is limited, and the model can not distinguish them very well. Further more, this unsupervised algorithm cannot use the overdue information effectively because of the absence of abnormal labels for transactions. MILAD are designed for solving the anomaly detection problems in these applications similar to the credit card risk control study.

---

> ### Comment · Reviewer_nXvi · 2023-11-22
>
> Thanks to the authors for addressing my comments

---

### Official Review · Reviewer_vHLH · 2023-11-01

**Soundness:** 2 fair
**Presentation:** 3 good
**Contribution:** 2 fair
**Rating:** 3
**Confidence:** 3

**Summary:**

The paper proposes a technique to find anomalies in credit card transactions using multi-instance learning.

**Strengths:**

The algorithm is suitable for financial fraud detection

**Weaknesses:**

1. The technique, as presented, is very narrow in scope, limited to financial / credit card data. To demonstrate wider applicability, more datasets and base line algorithms are required. Currently only one dataset and only one baseline algorithm has been used for evaluation which is insufficient.


2. Line 129 "... model to make the whole model to be more auto adaptive ..." -- This statement is misleading. Only some parts of the system are built with neural nets. The supervised part later uses xgboost contradicting the 'whole model' part of the statement.


3. Figure 2, Figure 3: Should plot graphs of both MILAD and DAGMM on the same graph. The training loss is not very important.

**Questions:**

1. Line 105 -- it is not clear what kind of architecture the transformer 'T' has -- encoder/decoder, etc.


2. It is not clear whether the attention network W is based on self-attention. For self-attention, a position encoding is also used. Does the position encoding apply to W if the attention is only among random samples?


3. Line 111-118 -- Under the Collective Assumption, are the samples in a bag not randomly selected? Since in this case we need to use self-attention, it might imply a dependency on sequence or the the samples being related in some manner. Else, how would we apply self-attention to a random collection of samples that are not in any guaranteed order?

---

> ### Author Response · Authors · 2023-11-16
>
> To questions:
> 1. The transformer network T is not encoder/decoder structure. The details of the transformer network T is introduced in the Appendix A.0.1;
> 2. Yes, the network W is based on the self attention. There is no random sample procedure.
> 3. The samples in a bag are not randomly selected.

---

> > ### Comment · Reviewer_vHLH · 2023-11-23
> >
> > Thanks to the authors for answering my questions.

---

### Meta-Review · Area_Chair_2Huz · 2023-12-15

**Metareview:**

The paper studies credit card default, and designed a multi-instance anomaly detection method. However, as pointed out by reviewers, the experiments are somewhat lacking; and the novelty of the proposed approach is unclear.

**Justification For Why Not Higher Score:**

The paper is clearly lacking in several aspects including novelty.

**Justification For Why Not Lower Score:**

N/A

---

### Decision · Program_Chairs · 2024-01-16

Reject